# Oncological Benefit versus Cardiovascular Risk in Breast Cancer Patients Treated with Modern Radiotherapy

**DOI:** 10.3390/jcm11133889

**Published:** 2022-07-04

**Authors:** Francisco Acevedo, Teresa Ip, María Orellana, Gonzalo Martínez, Luigi Gabrielli, Marcelo Andia, Cecilia Besa, Mauricio P. Pinto, Cesar Sánchez, Tomas Merino

**Affiliations:** 1Department of Hematology-Oncology, School of Medicine, Pontificia Universidad Católica de Chile, Santiago 8330077, Chile; fnacevedo@gmail.com (F.A.); mporella@uc.cl (M.O.); mauricio_pinto@outlook.com (M.P.P.); cgsanche@uc.cl (C.S.); 2Maule Health Service, Talca Hospital, Talca 3460001, Chile; tere.ip@gmail.com; 3Division of Cardiovascular Diseases, School of Medicine, Pontificia Universidad Católica de Chile, Santiago 8330077, Chile; doctorgonzalomartinez@gmail.com (G.M.); lgabriel@uc.cl (L.G.); 4Advanced Center for Chronic Diseases (ACCDiS), Santiago 8380000, Chile; 5Department of Radiology, School of Medicine, Pontificia Universidad Católica de Chile, Marcoleta 377, Santiago 8320000, Chile; meandia@uc.cl (M.A.); cbesa@uc.cl (C.B.); 6Millennium Institute for Intelligent Healthcare & Engineering (ANID), Vicuña Mackenna 4860, Macul, Santiago 7820436, Chile

**Keywords:** radiotherapy, breast cancer, cardio-oncology, cardiovascular disease

## Abstract

Radiotherapy (RT) is an essential part of breast cancer (BC) treatments. Unfortunately, heart exposure to radiation can also impair the long-term survival of patients. Our study aimed to quantify the oncological benefit and the cardiovascular (CV) risk associated with modern RT in a real-world cohort of BC patients. Our descriptive study enrolled BC patients who received adjuvant RT. Ten-year overall survival (OS) was estimated using Predict^®^ version 2.1 (National Health Service, London, UK). The basal risk of CV events was estimated using the American Heart Association (ACC/AHA) CV score. Treatment volumes and mean cardiac doses were obtained from RT treatment plan records. The increased risk of CV events due to RT was estimated using a model proposed by Darby. The risk of acute myocardial infarction or stroke mortality was estimated using HeartScore^®^ (European Society of Cardiology, Brussels, Belgium). A total of 256 BC patients were included in the study. The average age of patients was 57 years old (range: 25–91); 49.6% had left BC. The mean cardiac dose was 166 cGy (interquartile range (IQR) 94–273); the estimated hazard ratio (HR) for CV disease was HR 1.12 (confidence interval (CI) 1.04–1.24). The estimated baseline 10-year CV risk was 5.6% (0.2 to 51.2); CV risk increased by 0.9% (range 0.02–35.47%) after RT. The absolute risk of 10-year mortality from CV disease was 2.5% (0.1–9); RT was associated with an estimated 4.9% survival benefit (3.73–6.07) against BC death and a 0.23% (0.17–0.29) estimated increase in CV mortality. Modern RT decreased 10-year BC mortality by 4% but increased CV mortality by 0.2% in this cohort. Our findings encourage the implementation of personalized adjuvant RT treatments that balance risks and benefits to improve long-term BC patient survival.

## 1. Introduction

Globally, cardiovascular diseases (CVDs) and cancer are the first and second leading causes of death, respectively [1]. While CVDs are responsible for approximately 17.9 million deaths a year, accounting for 31% of worldwide deaths, cancer causes >10 million deaths every year. Among malignancies, recent studies indicate that female breast cancer (BC) is the most frequently diagnosed type of cancer worldwide, with >2 million cases a year [2]. In the United States, estimates indicate that 1/3.3 of women’s deaths are caused by CVDs, while 1/31.5 are related to BC [3]. Recently, major advances in BC diagnosis and treatment have translated into improved prognoses, reaching >95% survival in early-stage patients. Unfortunately, several treatments are also associated with side effects and toxicities (either acute or chronic) that may affect patients’ quality of life and even survival.

Previous studies have determined that several factors may be associated with cardiovascular (CV) risk in BC survivors. These include modifiable and non-modifiable factors, such as age, family history, obesity, alcohol use, and hormone replacement therapy. For example, it is well documented that oncological treatments are associated with cardiotoxic effects that increase CV risk in cancer survivors. A large study that collected real-world data from BC female patients assessed the rate of BC-specific death (BCSD) and non-BCSD at 10 years and determined that 14.17% of mortality could be attributed to BC, whereas 5.39% was due to CVDs [4].

Radiation therapy or radiotherapy (RT) is a key part of the standard treatment for the management of BC; RT consists of using aimed ionizing radiation to destroy cancer cells. In patients, RT prevents cancer recurrence after tumor removal [5,6]. Ionizing radiation damages the DNA of cancer cells within tumors and generates free radicals that eventually cause cell death. More recently, studies have suggested that the stromal microenvironment of tumors participates in the radiation response [7].

As pointed out, RT is associated with marked benefits in local disease control and patient survival [5] but also induces long-term CV damage that may manifest decades after the initial treatment, increasing CV-related mortality [8]. Recently, some of the mechanisms by which radiation causes cardiovascular damage have been elucidated [9]; this has allowed the development of novel RT techniques to protect the heart, as well as more sensitive diagnostic tools, such as two-dimensional speckle-tracking echocardiography. This technique detects subclinical disease after early treatments and, therefore, can prevent further damage [10].

Although several studies have compared dosimetry in a variety of RT techniques, the relative impact of the specific treatments and their relationships with baseline CV risk or potential oncological impacts have been poorly explored in the real-world setting.

Herein, we propose that treatment schemes for BC patients should be accompanied by an accurate assessment of individual CV risk. Thus, our study sought to quantify the risks versus the benefits for BC patients undergoing RT treatment by assessing their individual oncological/CV risks.

## 2. Materials and Methods

In this descriptive cross-sectional study, BC women ≥18 years old who received adjuvant RT with a conformational technique between May 2017 and August 2018 at the Cancer Center of the Pontificia Universidad Católica de Chile, Santiago, Chile, were analyzed. We included women with total or partial mastectomies, with or without chemotherapy, with or without hormone therapy, treated with RT to the chest wall/whole breast, with or without a boost to the tumor bed, and independent of the dose and fractionation. Women with a history of cancer other than BC, previous thoracic RT cardiac surgeries, and prior major cardiovascular pathologies were excluded. The first stage of the study included all BC patients who received RT with curative intent. The second stage excluded patients with missing or incomplete information that was needed to calculate the risk of recurrence using Predict or the risk of CV mortality using AHA criteria. A patient flow chart summarizing inclusion and exclusion is shown in Appendix A.

We calculated the 10-year estimated BC overall survival (OS) using the online tool Predict^®^ version 2.1 [11] (https://breast.predict.nhs.uk/tool, accessed on 14 December 2021) Centre for Mathematical Sciences, Wilberforce Road, Cambridge, UK CB3 0WA) and the risk of a CV event using the American Heart Association (AHA) Predictive Score [12], which has been previously validated in Chile [13].

All patients were treated in the supine position. Treatment volumes and mean cardiac doses were obtained from RT treatment plan records. Treatment volumes were delimited according to the EORTC consensus [14] and cardiac volumes according to the University of Michigan Cardiac Atlas recommendations [15]. Treatment was 3D conformal with the field-in-field technique to improve the homogeneity and coverage of the target. The breath-hold technique was not available. Treatment planning was performed using Eclipse version 8.6 (Varian Medical Systems, Palo Alto, CA, USA) and calculated with Pencil Beam Convolution Algorithm version 8.6.15.

The increased risk of CV events due to RT was estimated according to the model proposed by Darby [16]. In brief, this model was developed using a large cohort of Swedish and Danish patients treated for breast cancer and showed a linear relative increase of 7.4% per each median cardiac dose in gray. The increase in mortality from acute myocardial infarction or stroke was calculated according to the HeartScore^®^ of the European Association of Preventive Cardiology [17] (www.heartscore.org, accessed on 14 December 2021). This score was developed using 12 European cohort studies (*n* = 205,178) covering a wide geographic spread of countries with different levels of cardiovascular risk; it contains more than 3 million person-years of observation and 7934 fatal cardiovascular events. HeartScore^®^ estimates the risk of cardiovascular death based on age, sex, smoking habit, blood pressure, and blood cholesterol or total/HDL cholesterol ratio.

Dosimetry was analyzed according to the fractionation regimen (standard fractionation, moderate hypofractionation, accelerated partial fractionation (APBI), hyperfractionation, and palliative fractionation) and according to the target volumes treated (right/left breast and the inclusion/exclusion of nodal regions and internal mammary regions).

We estimated the number needed to treat (NNT) to avoid one breast cancer death at 10 years using the estimated breast cancer mortality tool Predict^®^ version 2.1 [11]. The absolute reduction in BC mortality due to adjuvant RT was estimated by multiplying the basal breast cancer mortality risk by 20%, based on the results of a previously published meta-analysis [5,6]. We estimated the number of patients treated with this radiotherapy plan required to increase cardiac mortality at 10 years by one (the number needed to harm, NNH) by multiplying the basal 10-year cardiovascular mortality risk from www.heartscore.org, accessed on 14 December 2021) by the estimated incremental mortality risk. We estimated the ratio of incremental survival attributable to RT versus the incremental CV mortality by dividing incremental survival by incremental cardiovascular mortality for each patient in the cohort and by doing the same for specific subgroups (left vs. right, under vs. over 60 years old). We used HR with 95% CI from the Darby model to estimate the incremental risk from pre-radiotherapy absolute risk.

Since the characteristics of normal distribution were not met, non-parametric statistics were used. For the continuous variables, we used the median and interquartile ranges (Q1–Q3), and *p* < 0.05 was considered significant. To evaluate the median differences (heart dose and the CV survival–death relationship), we used the Kruskal–Wallis test or ANOVA for variables with multiple groups. For multivariate analysis, we used ANOVA.

The Scientific Ethics Committee at the Pontificia Universidad Católica de Chile approved this project; the assigned identification number was 1805100.1

## 3. Results

A total of 256 patients were included in our study. The main patient characteristics are summarized in Table 1. The median age at diagnosis was 57 years old (range: 25–91). The most frequent comorbidity was hypertension (HT; *n* = 86; 33.6%), followed by dyslipidemia (*n* = 31; 12.1%) and type 2 diabetes (*n* = 15; 5.8%). The median body mass index (BMI) was 26.45 kg/m^2^ (range: 18.7–44.2). Almost half of the participants (*n* = 127; 49.6%) had BC that affected their left breast, and 80.8% (*n* = 240) were invasive ductal carcinomas. Most tumors were T1 (*n* = 139; 46.8%), with the number of patients at various nodal stages as follows: N0 136 (62%), N1 53 (24%), and N2-N3 28 (12%). Regarding treatments, 25.9% received neoadjuvant chemotherapy that consisted of adriamycin cyclophosphamide (four cycles) and weekly taxanes (12 weeks); 70.3% of patients had a partial mastectomy. In addition, 65.3% (*n* = 194) received breast RT, 22.2% (*n* = 66) had chest wall RT, 42% (*n* = 125) had supraclavicular RT, and 9% (27) had RT to the internal mammary nodes.

Initially, we sought to estimate the relative increase in CV risk derived from RT treatments. The dose to the whole heart was 166 cGy (IQR 94–273). These values correspond to a 12% increase in relative CV risk (HR 1.12; CI 95%; 1.04–1.24) within the next 20 years after treatment (Figure 1A). For an average 60-year-old patient with a basal 10-year CV mortality risk of 2% and a mean cardiac dose of 166 cGy, the estimated incremental cardiovascular mortality is 0.13%; for the first and last quartiles, dose values are 0.08% and 0.22%, respectively.

Cardiac dosimetry and the corresponding CV risk varied significantly according to the RT plan (Table 2). RT doses for the left and right breasts were 245 (174–342) cGy and 99 (77–161) cGy, respectively; the relative increases in CV risk were 18.1% (CI 95%: 5.9–35.5) vs. 7.3% (CI 95%: 2.4–14.3); (*p* < 0.001). Fractionation was also associated with the dose. For standard fractionation, moderate hypofractionation, APBI, hyperfractionation, and palliative radiotherapy, the corresponding dose (cGy) means (P25, P75) and HR values were 215 (121–354) and HR 1.16 (1.05–1.31), 166 (91.269) and HR 1.12 (1.04–1.23), 123 (96–163) and HR 1.09 (1.03–1.17), 283 (91–720) and HR 1.2 (1.08–1.39), and 193 (67–723) and HR 1.14 (1.05–1.27), respectively. With significantly lower cardiac doses, there was a correspondingly lower cardiovascular risk for moderate hypofractionation and APBI, but the significantly higher doses for hyperfractionation and palliative radiotherapy schemes yielded higher risks (*p* = 0.01, *p* = 0.001, *p* = 0.005, and *p* = 0.012, respectively).

Furthermore, the irradiation of nodal areas was also associated with higher doses and increased CV risk: values were 199cGy (102–322) and HR 1.14 (1.05–1.28) for the axillary group three node region and 418 cGy (225–531) HR 1.3 (1.12–1.6) for the internal mammary node chain. The doses and CV risk for patients not treated in these areas were 160 cGy (91–219) and HR 1.12 (1.04–1.23) and 163 cGy (1.04–1.23) and HR 1.14 (1.04–1.23) (*p* = 0.004 and *p* = 0.001, respectively). The use of a boost to the tumor bed was associated with an increased dose to the heart of 193 cGy (121.305), *p* = 0.0006. Multivariable analysis confirmed that hyperfractionated radiotherapy, left breast treatment, and internal mammary nodes were associated with increased cardiac doses, all with values of *p* < 0.001, but other fractionation regimens, the treatment of SCV nodes, and the use of a boost were not.

Next, we estimated the absolute mortality risk for the subset of 71 patients who had complete datasets (Figure 1B). Thirty-two (44%) and forty (56%) patients had right and left BC, respectively, within this subset. The mean dose to the chest wall/breast (CI 95%) was 4275 cGy (3000–5000), and the median cardiac dose (CI 95%) was 166 cGy (94–273). The estimated baseline 10-year CV risk was 5.6% (0.2 to 51.2); 40 patients (40.8%) were classified as high-risk according to AHA definitions (>7.5% at 10 years), while 26 patients (36.6%) had >10% risk at 10 years. With RT, the estimated 10-year CV risk increased by 0.9% (range 0.02–35.47%). The 10-year risk of death from CVD was 2.5% (0.1–9). Then, we estimated the risk of death by BC versus the benefit derived from RT. First, the 10 year-OS with surgery only (without adjuvant treatments) was 70% (1–96) at 10 years. Adjuvant systemic treatments increased 10-year OS to 91% (32–99). Thus, the expected benefit of adjuvant RT was 4.9% (0.25–22.25). Finally, we estimated the ratio of survival attributable to RT versus CV mortality. As explained, RT was associated with a 4.9% survival benefit, and this corresponds to a number needed to treat (NNT) of 20 (16–26) to prevent one death from BC in this population. The increase in the absolute risk of CV mortality was 0.23% (0.17–0.29), and this corresponds to a number needed to harm (NNH) for CV mortality associated with RT of 434 (344–588). The ratio of increased OS associated with RT to BC/CV mortality was 22.8 (9.78), meaning the estimated increase in BC survival probability was 22 times greater compared to the risk of CV mortality in this cohort. By subgroup, for patients with left breast and right breast cancer, these numbers were 12.2 (7.40) and 37.3 (14.108), respectively, *p* = 0.18; for <60-year-old and ≥60-year-old patients, these values were 79 (18.206) and 14.8 (7.37), respectively, *p* = 0.001.

## 4. Discussion

Early reports on radiation-induced heart disease were published in the 1970s. In the following decades, several studies confirmed an increase in CV mortality associated with RT, specifically among BC patients [8]. Furthermore, case-control studies demonstrated that treatment-derived radiation exposure increased the risk of ischemic heart disease in a dose-dependent manner. Our work sought to estimate the risk of long-term CV complications associated with a variety of modern RT treatment schemes commonly used in BC patients, including different fractionations and volumes treated, based on dosimetry algorithms routinely used in current clinical practice. Remarkably, we confirm a reduction in radiation dose from 13.3 Gy, commonly applied in the 1970s, to the modern dose of 1.7 Gy. Modern RT techniques maintain effectiveness while decreasing cardiac exposure. Evidently, the therapeutic potential of modern RT must be balanced with the associated risks, especially given the major advances in systemic treatments for BC patients.

In line with previous studies, our results confirm significant differences in cardiac dosimetry and CV risk according to BC laterality, fractionation, and irradiation of the nodal areas [18]. Overall, we estimate a 4.8% benefit in 10-year OS derived from adjuvant RT in our cohort. Similarly, previous reports based on the SEER and National Cancer databases have reported benefit values that ranged between 2.9% and 3.8% [15]. In contrast, a study based on a national Japanese BC registry reported no benefit from modern post-mastectomy/NACT RT in patients who had fewer than four positive residual nodes. Despite this, investigators concluded that modern RT may still be beneficial for patients with more than four positive residual nodes, and they called for randomized trials to test this hypothesis [19].

The estimated 10-year risk for CV events and the baseline CV mortality in our cohort were 5.6% and 2.5%, respectively. The first can be attributed to patient comorbidities; 40% of participants in our cohort were categorized as having high CV risk according to the AHA criteria. As demonstrated by others, RT causes a 6.4% increase in the 10-year risk of CV events in these patients [20,21]. Secondly, the baseline CV mortality levels in our cohort were also similar to previous reports that indicate 1–1.3% and 2–3.4% CV mortality in patients with right and left BC, respectively [22,23,24,25]. Although our results suggest that the benefit derived from RT surpasses the associated risk of CV mortality, we should also keep in mind patient heterogeneity. Thus, for some patients, RT may not offer a benefit, particularly for those at high CV risk. In this regard, we conducted an exploratory analysis using (www.heartscore.org, accessed on 14 December 2021) [17] to estimate the remaining CV if optimal control of modifiable CV risk factors had been achieved, for example, optimal blood pressure control in hypertensive patients or smoking cessation in smokers. In the above-mentioned situation, the CV risk would be reduced by 30%, emphasizing the importance of personalized RT treatments. We hypothesize that tailored RT treatments should involve assessments of CV/oncological risk to identify individuals more likely to benefit from RT and to elaborate optimized plans for high-CV-risk patients. This approach also calls for the implementation of a multidisciplinary cardio-oncology team to manage patient comorbidities and risk factors, balancing benefits and risks for each case. A recent review article discussed the possibilities of personalized RT for BC patients, focusing on biomarkers and predictors of response, including tumor subtypes and genomic analyses; however, other patient-related factors, such as lifestyle, CV risk, and comorbidities, were not discussed [26].

Evidently, the elaboration of tailored or personalized RT plans is a complex process that demands the evaluation of several factors. Figure 2A shows a diagram that summarizes our proposal. Initially, patients are divided according to their relative risk of BC recurrence. A small subset is categorized as having a very low risk of recurrence (Figure 2A, left), including patients >60 years old, with T1/N0 cancer, with estrogen receptor (ER)+, with HER2-, and with a low histological grade. For these patients, we propose considering modifications in the RT plan to reduce CV risk. This can be achieved by modifying the target volume, for example, by using partial breast RT, controlling the respiratory cycle to treat the patient in a deep-breath hold, or modifying the position, such as using the prone position for large-breast patients or applying a different technique utilizing modern VMAT, IMRT, or photon-RT at certain specialized cancer centers. Conversely, a proportion of patients are categorized as having a very high risk of recurrence, including patients <60 years old, with T1-T4/N2-N3 cancer, with ER-, with HER2+, and with a high histological grade (Figure 2A, right). For these patients, we propose proceeding with the optimal oncological RT plan. Lastly, most BC patients fall into the category of “low–medium–high” risk of recurrence (Figure 2A, center). In this group, we propose an estimation of the oncological gain versus the CV death (OG/CVD) ratio. As shown in the diagram in Figure 2A, we postulate a cutoff value of OG/CVD = 3 to discriminate between optimal oncological RT (OG/CVD ≥ 3) or reconsideration/modification of the RT plan (OG/CVD < 3). In summary, a gain/risk-adjusted and tailored RT plan should balance the BC recurrence risk and the CV risk of patients, given that lower or higher RT doses can increase the BC recurrence risk or the CV risk.

Some of the limitations of our study include the cross-sectional design and a relatively small sample size that may not be representative of a wider BC population. In addition, although some of these predictors have been extensively validated, their applicability is limited by recent changes in patient management of CV and cancer risks. Additionally, RT-associated CV risk models are based on simple estimates of dosimetry. To date, the radiosensitivity of specific cardiac substructures remains unclear, and this means that the estimations of CV risk are still inaccurate [27]. New score models have recently been presented and validated in the Asian population, showing a dose-dependent relation between the dose of anthracyclines and CV events, but unfortunately, this factor was not part of our analysis. Consequently, we simply assumed a linear dose–risk relationship. This model was based on standard tangent field breast RT and may not predict the CV risk of modern state-of-the-art techniques, such as intensity-modulated radiotherapy (IMRT) and volumetric arc therapy (VMAT). On the other hand, >50% of patients in our cohort received systemic treatments that may have a synergistic effect on cardiotoxicity. Unfortunately, our model does not account for these effects; therefore, this is a limitation of our study. Finally, although we report patient characteristics and comorbidities, we were unable to estimate the potential confounding effect of these factors on RT variables. Even with all these limitations, we believe that a significant proportion of BC patients could benefit from cardiac-sparing RT techniques, such as partial breast RT for low-risk patients and deep-breath-hold RT for higher-risk, left-sided BC patients who require nodal irradiation.

## 5. Conclusions

In our study, modern RT was associated with a 4% decrease in 10-year BC mortality and a 0.23% increase in CV mortality. The use of hyperfractionated schemes and the treatment of the left breast and internal mammary chain were associated with greater cardiac exposure. Meanwhile, age under 60 years was associated with a higher oncological survival benefit/cardiovascular risk ratio. Although these values must be analyzed individually, our findings confirm the importance of optimal RT treatment plans with cardiac protection and the effective management of comorbidities to improve BC patient survival.

## Figures and Tables

**Figure 1 jcm-11-03889-f001:**
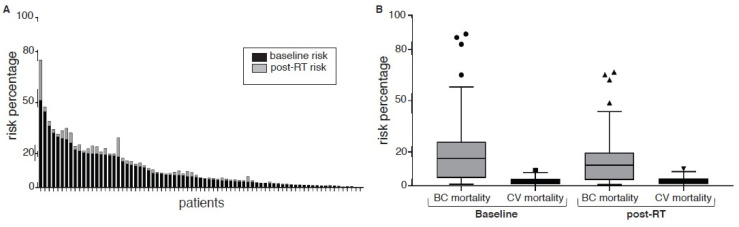
(**A**) Baseline and post-RT 10-year major cardiovascular event risk. (**B**) Baseline and post-RT 10-year breast cancer and cardiovascular risk mortality. Circles show baselines values, triangles are post-RT.

**Figure 2 jcm-11-03889-f002:**
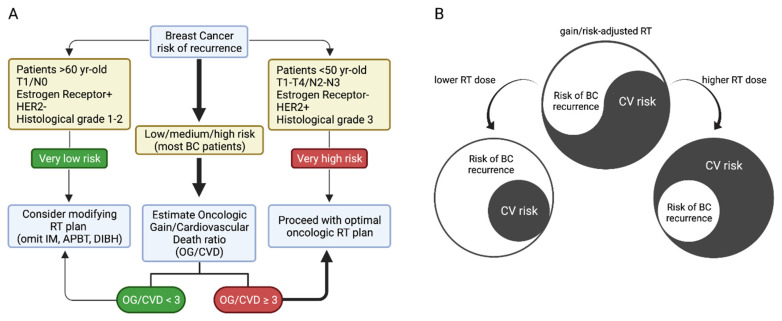
(**A**) Proposal for the classification of patients with indications of radiotherapy for breast cancer. (**B**) simplified scheme of cardiac dose impact on risk balance.

**Table 1 jcm-11-03889-t001:** Baseline patient characteristics.

Variable; Total *n* (Units)	Value (Range)
Mean age; *n* = 256 (yr)	57 (5–91)
Mean weight; *n* = 167 (kg)	69 (46–109)
Mean height; *n* = 136 (cm)	158 (140–177)
Variable	*n* (%)
TNM stage	
T1	121(47.2)
T2	83 (32.2)
T3	15 (5.8)
T4	32 (12.5)
Unknown	5 (1.9)
N0	136 (53.1)
N1	53 (20.7)
N2	21(8.2)
N3	7 (2.7)
Unknown	39 (15.2)
BMI; *n* = 136	
Normal weight (18.5–24.9 kg/m^2^)	45 (33)
Overweight (25–29.9 kg/m^2^)	49 (36)
Obese (>30 kg/m^2^)	42 (31)
Comorbidities; *n* = 256	
None	95 (37.1)
Hypertension	86 (33.6)
Dyslipidemia	31 (12.1)
Coronary Cardiopathy	3 (1.2)
DMNIR	8 (3.1)
DMIR	7 (2.7)
Current smoker	
Systemic treatment; *n* = 129	
Adjuvant chemotherapy	63 (48.8)
Neoadjuvant Chemotherapy	64 (49.6)
Trastuzumab	6 (4.6)

DMNIR: Diabetes mellitus non-insulin required; DMIR: Diabetes mellitus insulin required.

**Table 2 jcm-11-03889-t002:** Treatment volume, fractionation, and corresponding cardiac dose.

Variable	*n*	Median * (cGy)	IQR (Q1/Q3)
Global	257	166.0	94/273
Right breast	127	99.0	77/161
Left breast	130	245.0	174/342
Fractionation			
STD	64	215	121/354
Moderate hypofractionation	149	166	91/269
Accelerated partial breast	31	123	96/163
Hyperfractionation	4	283	91/720
Palliative scheme	8	193	67/723
Nodal region			
With SCV	105	199.0	102/332
Without SCV	152	160.0	91/219
With IM	18	418.5	225/531
Without IM	239	163.0	92/246
Boost			
Yes	108	193.5	121/305
No	149	143.0	86/227

* Mean cardiac dose (cGy); IQR: interquartile range; SCV: supraclavicular; IM: internal mammary.

## Data Availability

Not applicable.

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
