# Peer review of "Oncological Benefit versus Cardiovascular Risk in Breast Cancer Patients Treated with Modern Radiotherapy"

_jcm, 2022, doi:10.3390/jcm11133889_

Round 1

Reviewer 1 Report

The authors tackle an important topic on breast radiation and its effect on cardiovascular toxicity and mortality. Despite the importance of this topic, the information presented are known with minimal novelty added to the literature. 

The manuscript can be strengthened if the following are addressed:

- Authors should confirm that all patients were treated in supine position. 

- How many patients got cardiotoxic systemic therapy (e.g doxorubicin, Trastuzumab, etc.)? This can bias the RT toxicity rates. I would suggest looking into this and stratifying patients' CV into RT + cardiotoxic systemic therapy vs RT alone.

- Per authors methods section, all patients were treated with 3D technique even those nodal irradiation. Nowadays, a lot of institutions utilize IMRT and VMAT to decrease toxicity, in addition to utilizing prone techniques or deep inspiration breath holds. The authors minimally tackled these important topics and should dwell more on that in their discussion and limitations. Authors suggest personalized selection of patients but do not offer recommendations on those techniques that can minimize the heart dose.

- In the abstract line 28. I recommend the authors to reword this sentence as it might be mis-read. Right now it sounds as modern RT is increasing CV mortality compared to older techniques. 

- Line 54, "RT uses high energy". The word ionizing radiation or photons should be added to it.

Author Response

We would like to thank the editor and the reviewers for considering our manuscript and for taking the time to review our work. We have carefully read all comments and requests and have prepared point-by point replies. We hope this new revised and improved version of our manuscript is suitable for publication

Reviewer 1

The authors tackle an important topic on breast radiation and its effect on cardiovascular toxicity and mortality. Despite the importance of this topic, the information presented are known with minimal novelty added to the literature. 

The manuscript can be strengthened if the following are addressed:

1- Authors should confirm that all patients were treated in supine position. 

R.1- We confirm all patients were treated in supine position. This sentence has been added to the methods section (LINE 99)

2- How many patients got cardiotoxic systemic therapy (e.g doxorubicin, Trastuzumab, etc.)? This can bias the RT toxicity rates. I would suggest looking into this and stratifying patients' CV into RT + cardiotoxic systemic therapy vs RT alone.

R.2- We agree with the reviewer in this point. Table 1 shows that 129 out of 256 patients received cardiotoxic systemic treatment, and 123 of them had Anthracycline chemotherapy and 6 had Trastuzumab treatment. As the reviewer points out additional cardiotoxic treatment(s) could potentially have a synergistic effect upon patient cardiotoxicity. Unfortunately, our model does not account for this effect and therefore has been added as a limitation of our study at the end of the discussion section (starting at LINE 302).

3- Per authors methods section, all patients were treated with 3D technique even those nodal irradiation. Nowadays, a lot of institutions utilize IMRT and VMAT to decrease toxicity, in addition to utilizing prone techniques or deep inspiration breath holds. The authors minimally tackled these important topics and should dwell more on that in their discussion and limitations. Authors suggest personalized selection of patients but do not offer recommendations on those techniques that can minimize the heart dose.

R.3- In response to the request by the reviewer we have added a paragraph into the discussion section offering alternatives to minimize heart doses. These are mainly related to modifications in target volumes by using partial breast RT or modifications in patient positioning (Starting at LINE 275)

4- In the abstract line 28. I recommend the authors to reword this sentence as it might be mis-read. Right now it sounds as modern RT is increasing CV mortality compared to older techniques. 

R.4- As requested by the reviewer we have rephrased this sentence in the revised version of our manuscript.

5- Line 54, "RT uses high energy". The word ionizing radiation or photons should be added to it.

R.5- As requested we have replaced “RT uses high energy” for “RT consists on ionizing radiation”

Reviewer 2 Report

In the present article, the authors evaluate the effect of radiation on cancer regression and off-target effects on cardiovascular health in patients. I have several reservations, my comments are appended as below:

1. Radiotherapy- describe the dose quantitatively. 

2. Patient selection- describe inclusion and inclusion criterion, preferably provide a flow chart.

3. Chemotherapy- describe the type of treatment. 

4. Do authors have background information on patients, whether they have related ailment as blood pressure/hypertension?

5. Do authors estimate PFS/OS?

6. Primarily, patients are subjected to localized radiation, and how is it supposed to directly impact cardiovascular health?

7. Do authors observe additional cofounders as BMI among the patients?

Author Response

We would like to thank the editor and the reviewers for considering our manuscript and for taking the time to review our work. We have carefully read all comments and requests and have prepared point-by point replies. We hope this new revised and improved version of our manuscript is suitable for publication

Reviewer 2

In the present article, the authors evaluate the effect of radiation on cancer regression and off-target effects on cardiovascular health in patients. I have several reservations, my comments are appended as below:

  1. Radiotherapy- describe the dose quantitatively. 

R.1 We are unsure about this comment. Table 1 of our manuscript shows treatment volumes and fractionation and corresponding cardiac dosing of participants

  1. Patient selection- describe inclusion and inclusion criterion, preferably provide a flow chart.

R.2 We have added a brief description of inclusion and exclusion criteria in the methods section as requested (LINE 90). Also we are adding a new supplementary flow chart describing the inclusion and exclusion of patients in our study (supplementary fig. S1)

  1. Chemotherapy- describe the type of treatment.

R.3 We used Adriamycin Cyclophosphamide for 4 cycles and weekly Taxane for 12 weeks (LINE 151).

  1. Do authors have background information on patients, whether they have related ailment as blood pressure/hypertension?

R.4 Yes. Patient background information is summarized in Table 1, including BMI and comorbidities such as hypertension, dyslipidemia, and type-2 diabetes.

  1. Do authors estimate PFS/OS?

R.5 unfortunately, since this is a cross-sectional study we do not have clinical follow up data on these patients and PFS/OS cannot be calculated. However, using Predict we estimated patients’ 10-year progression-free survival and 10-year overall survival. These values were: 10-yr PFS=70%+/-15 and 10-yr OS=91%+/-8.7

  1. Primarily, patients are subjected to localized radiation, and how is it supposed to directly impact cardiovascular health?

R.6- it is well documented that even with modern radiotherapy scattered radiation at lower doses is deposited in cardiac substructures, especially the left ventricle and the left anterior descending artery. Over time, this can lead to cardiac remodeling and accelerated arteriosclerosis mediated by an inflammatory response. Currently, this is a very active area of research that continues to evolve with interesting findings that could potentially translate into better cardioprotective measures for patients in the near future.  

  1. Do authors observe additional cofounders as BMI among the patients?

R.7- this is a very interesting point. Certainly, several other comorbidities or conditions could potentially have an impact on radiotherapy treatment variables. These include: BMI, hypertension, smoking and diabetes. This relevant point is another limitation of our study that has been added into the discussion section of the revised manuscript (Starting at LINE 302).

Round 2

Reviewer 1 Report

The authors tackled the comments and listed their limitations. The manuscript looks better now.

Reviewer 2 Report

All my concerns are addressed.